# Improving population scale statistical phasing with whole-genome sequencing data

Rick Wertenbroek[1,2]*, Robin J. Hofmeister[1], Ioannis Xenarios[1], Yann Thoma[2‡], Olivier Delaneau[3‡]*

1 University of Lausanne, Lausanne, Vaud, Switzerland, 2 School of Engineering and Management Vaud (HEIG-VD), HES-SO University of Applied Sciences and Arts Western Switzerland, Yverdon-les-Bains, Vaud, Switzerland, 3 Regeneron Genetics Center, Tarrytown, New York, United States of America

‡ These authors co-supervised on this work.
* rick.wertenbroek@unil.ch (RW); olivier.delaneau@regeneron.com (OD)

**Data Availability Statement:** The UK Biobank data set is a combination of Imaging, Genetics, Health linkages, Biomarkers, Activity monitors, Online questionnaires, and Samples from 500,000

## Abstract

Haplotype estimation, or phasing, has gained significant traction in large-scale projects due to its valuable contributions to population genetics, variant analysis, and the creation of reference panels for imputation and phasing of new samples. To scale with the growing number of samples, haplotype estimation methods designed for population scale rely on highly optimized statistical models to phase genotype data, and usually ignore read-level information. Statistical methods excel in resolving common variants, however, they still struggle at rare variants due to the lack of statistical information. In this study we introduce SAPPHIRE, a new method that leverages whole-genome sequencing data to enhance the precision of haplotype calls produced by statistical phasing. SAPPHIRE achieves this by refining haplotype estimates through the realignment of sequencing reads, particularly targeting low-confidence phase calls. Our findings demonstrate that SAPPHIRE significantly enhances the accuracy of haplotypes obtained from state of the art methods and also provides the subset of phase calls that are validated by sequencing reads. Finally, we show that our method scales to large data sets by its successful application to the extensive 3.6 Petabytes of sequencing data of the last UK Biobank 200,031 sample release.

## Author summary

Haplotype estimation, also known as phasing, is now applied to population scale projects, typically of hundreds of thousands of samples to millions of samples. Generally phasing relies on statistical methods as they provide very accurate results for common variations. However, for rare and very rare variants the lack of statistical power often results in poor phasing. The large amount of rare variations discovered with whole-genome sequencing as well as the number of samples makes it expensive to process. We have developed the SAPPHIRE method that leverages whole-genome sequencing data to verify and correct the phase at poorly phased variant loci. It does so by finding sequencing reads that contain both the poorly phased variant and an accurately phased common variant. SAPPHIRE scales with large data sets by specifically targeting variation where statistical phasing

participants, each with their own data sharing policies and restrictions. These restrictions do not allow the data set to be shared in a completely unrestricted way. However, readers may apply for access at https://www.ukbiobank.ac.uk/enable-your-research/apply-for-access The data and scripts to generate the figures of this paper are available in the supporting information files S1–S9 Data. The software developed for the SAPPHIRE method is open-source and available at https://github.com/rwk-unil/sapphire.

**Funding:** O.D. was supported by SNF grant number: SNSF-PP00P3_176977. https://www.snf.ch/en R. W. and Y. T. are supported by HEIG-VD. The funders did not play any role in the study design, data collection, and analysis, decision to publish, or preparation of the manuscript.

**Competing interests:** The authors have declared that no competing interests exist.

performed poorly, therefore it reduces the quantity of sequencing data to be processed and combines the advantages of both read-based and statistical approaches. We show the efficiency of SAPPHIRE by improving the estimated haplotypes for 200,031 samples in the UK Biobank. SAPPHIRE is free and available as open-source software.

## Introduction

In the era of biobanks, population-scale sequencing is becoming increasingly common [1], producing variant calls and comprehensive data sets that depict the genomic landscape across a vast number of samples [2]. To enable analyses at the haplotype level, it is necessary to phase the genotype data produced by sequencing. When dealing with data sets comprising thousands to millions of samples, statistical methods like SHAPEIT5 [3] or Beagle5 [4] are the standard approach. These methods borrow information across many samples in the population in order to produce precise haplotype estimates for common variants. Nevertheless, they tend to exhibit higher error rates when handling rare variants due to the limited information available for those in the population. Conversely, haplotype assembly methods, like WhatsHap [5] for instance, utilize local read reassembly techniques to group nearby variants into fully resolved haplotype blocks, often called phase sets. This type of approach offers the advantage of phasing variants with an accuracy level that remains independent of allele frequency, but typically provides only partial estimates, often spanning a few kilobases at most [5]. Indeed, variants located too far apart cannot be linked by the same set of reads and thus cannot be phased together. To harness the strengths of both approaches, a good strategy is to combine statistical phasing methods with haplotype assembly methods, as previously proposed in the phasing pipeline based on WhatsHap [5] and SHAPEIT4 [6]. While this approach leads to high accuracy levels, it faces limitations in scaling to the vast amount of samples present in modern sequencing data sets, comprising thousands or even millions of samples.

In this paper, we address this challenge and introduce an accurate and efficient haplotype alignment method, the Smart and Accurate Polishing of Phased Haplotypes Integrating Read Enhancements (SAPPHIRE) method. SAPPHIRE is primarily designed to refine the haplotypes estimated by SHAPEIT5 [3] but is applicable on any phased haplotypes. Our novel method capitalizes on the phasing confidence scores provided by SHAPEIT5 to pinpoint poorly phased rare variants. Then, it employs local read realignment techniques to correct errors at these poorly phased sites. This targeted approach leads to a substantial reduction in the volume of sequencing data that requires processing, reducing it by orders of magnitude. This makes it a viable solution even for extraordinarily large sequencing data sets. To illustrate its speed and accuracy, we applied SAPPHIRE to an extensive data set comprising 200,031 UK Biobank samples, each with whole-genome high-coverage sequencing, totaling more than 3.5 Petabytes of data. Our findings show that our method can traverse all this sequencing data at a relatively modest computational cost and achieve a significant enhancement of statistically estimated haplotypes, notably at rare variants and singletons.

## Results

### Overview of the SAPPHIRE method

Similar to long-read polishing [7–9], where errors in long-reads are corrected by aligning highly accurate short-reads, SAPPHIRE corrects phase errors in estimated haplotypes by aligning highly accurate short reads. The type of error introduced by statistical phasing can be

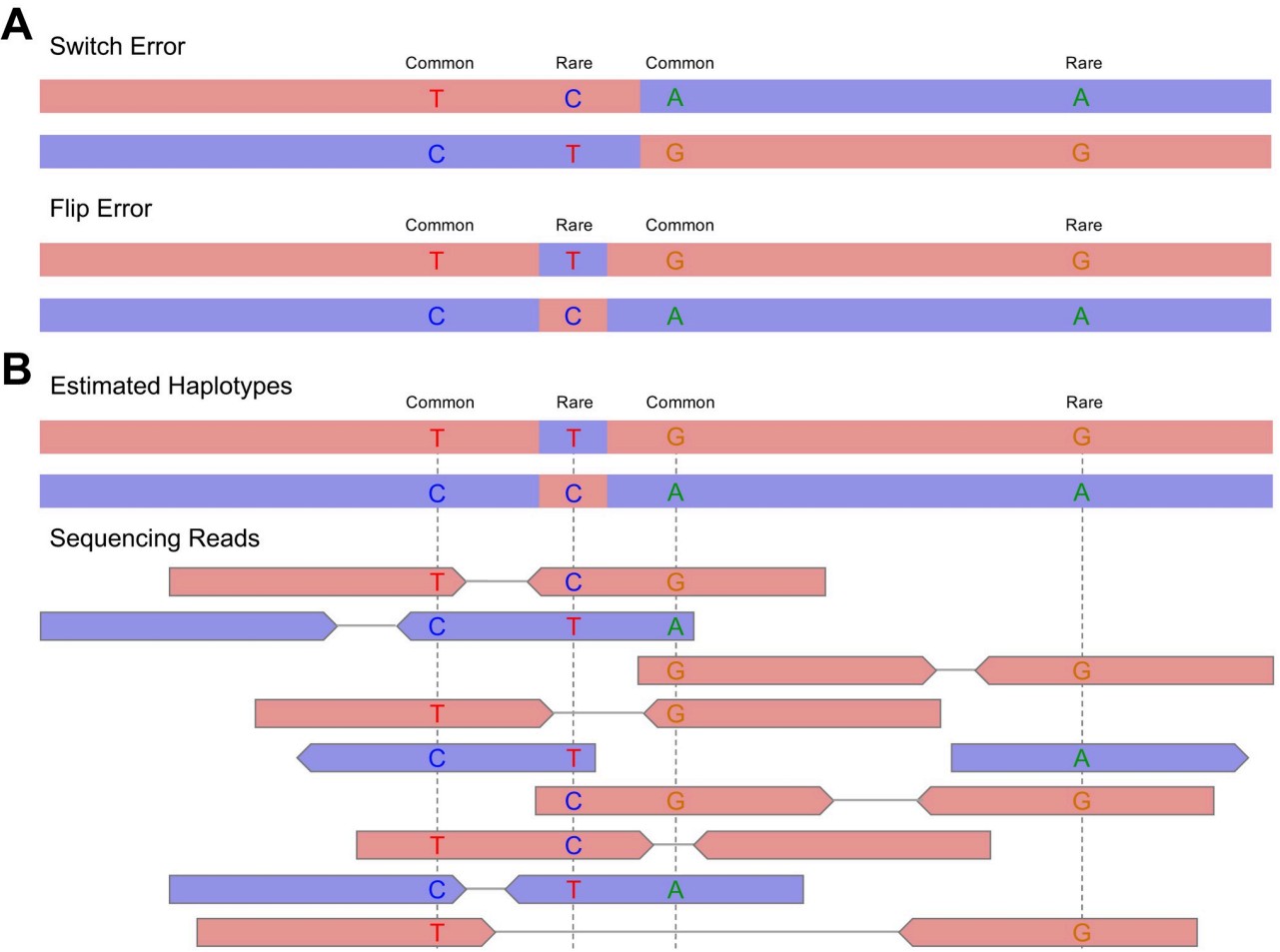

**Fig 1. Phasing errors and read-based validation of phase. A.** Phasing errors come in two types, switch errors where the entire contiguous segments of haplotypes are phased incorrectly and flip errors where only a single heterozygous genotype is flipped on a correctly phased haplotype background. **B.** Whole-genome sequencing reads aligned to the estimated haplotypes allow to validate or contradict the phase relationship of heterozygous variants that are in close enough proximity to be carried by a single sequencing read or a pair of sequencing reads. In this example we can see that the two common heterozygous variants have been phased correctly with regards to each other. However, the rare variant T/C was phased incorrectly (assigned to the wrong haplotype) and should therefore be switched. The rare variant G/A is phased correctly. The pink and blue colors are used to represent the two real haplotypes (maternal and paternal haplotypes). Mixed colors in estimated haplotypes indicate a phasing error.

classified into two categories, often dependent on allele frequency (see Fig 1A). At common variants, defined in [3] by a minor allele frequency (MAF) above 0.001 i.e., 0.1%, we mostly encounter switch errors, where entire contiguous segments of haplotypes are incorrectly phased. At rare variants, we mostly encounter flip errors, where only a single heterozygous genotype is flipped onto a correctly phased haplotype background. In large-scale sequencing data sets, such as the UK Biobank, the large number of samples being sequenced implies (i) very good haplotype estimates at common variants and thus few switch errors and (ii) an excess of rare variants which makes flip errors the primary source of error in this type of data sets. In this work, we propose an approach to quickly identify flip errors in large scale sequencing studies, and to correct these errors by performing local re-alignment of the overlapping sequencing reads (flip error detection and correction, Fig 1B). To achieve this, we first employ SHAPEIT5 for statistical phasing, as it provides a phasing confidence score for each rare heterozygous genotype. This score essentially corresponds to the probability of the phase the

software reports and enables us to pinpoint the subset of rare heterozygous genotypes with low phasing confidence. Then, we extract sequencing reads overlapping these poorly phased variants and use these reads to re-phase all these problematic variants, when possible. In practice, we consider each of these variants in turn and check if any of the extracted reads can connect the variant of interest to a nearby common heterozygous variant. By checking the allelic content of these phase-informative reads, we can collect evidence (number of reads) that confirm (validate) or invalidate the phase relationship reported by statistical phasing. Each time the phase relationship between a low confidence-phased heterozygous genotype to its neighbors is invalidated, we flip it. For high-confidence genotypes (including common variants), we do not make any adjustments, and assume they are correctly phased. With this method, even singletons (minor allele count of 1 in the population) can be phased if they are within read-pair distance of another nearby common variant. It is worth noting that this approach can also be used to refine the output of other statistical phasing methods, even though they do not provide phasing confidence scores. In such cases, we propose using allele frequency as a proxy for phasing confidence to identify possible phasing errors. The SAPPHIRE method details are described in the Methods section.

## SAPPHIRE on the UK biobank with whole-genome sequencing data

We applied the SAPPHIRE method described above on two data sets of the UK Biobank. (1) Whole-genome sequencing (WGS) data on chromosome 20 for 147,754 samples, a subset of the 150,119 samples release [10] with parental genomes of duos and trios excluded for validation. SAPPHIRE was applied on this data set, after a first statistical phasing pass through SHAPEIT5 as presented in [3]. (2) WGS data for all autosomes of the newest release of 200,031 samples also phased with SHAPEIT5 [11]. The data set of (1) was used to estimate the phasing accuracy using the offspring genomes as validation data. The data set of (2) includes the full set of samples (parental and offspring genomes) and was used to compute the costs of the SAPPHIRE method at scale, when applied genome-wide to WGS data for a large number of samples. In addition, we also used (2) to study the parental origin of de novo mutations, evaluate the number of genotypes that could be reassessed, and retrospectively evaluate the phasing confidence scores provided by SHAPEIT5.

## Phase quality of UK Biobank data after SAPPHIRE

The 147,754 sample data set was used to assess the phasing quality improvement after SAPPHIRE when applied on the haplotypes produced with Beagle5.4 [12] and SHAPEIT5 [3]. We used parent-offspring trios and duos ($N = 31$ and $N = 432$ respectively) to infer the true set of offspring haplotypes using Mendelian inheritance. With the resulting haplotypes as ground truth, we assess the accuracy of four phasing sets: statistical phasing performed with SHAPEIT5 and Beagle5.4, and with SAPPHIRE rephasing applied on both. The phasing quality is measured by computing the switch error rate (SER), which is the fraction of successive heterozygous genotypes phased incorrectly as defined in [3]. Because statistical phasing methods have a switch error rate that is correlated with the frequency of a variant, we measured switch error rates within bins of minor allele counts (MAC). Phasing performance is reported in Fig 2 and shows that the polished haplotypes exhibit lower phasing error rates at all minor allele count bins especially for rare variants. This is particularly apparent for singletons, where we go from 34% on the SHAPEIT5 haplotypes before SAPPHIRE to 21% after, a reduction of 13% in error rates. On the Beagle5.4 haplotypes we reduce the 50% SER to 32.5%, a reduction of 17.5% in error rates. Importantly, these results include all singletons, covered or not by phase-informative reads. When focusing on the subset of genotypes that can be re-phased using our

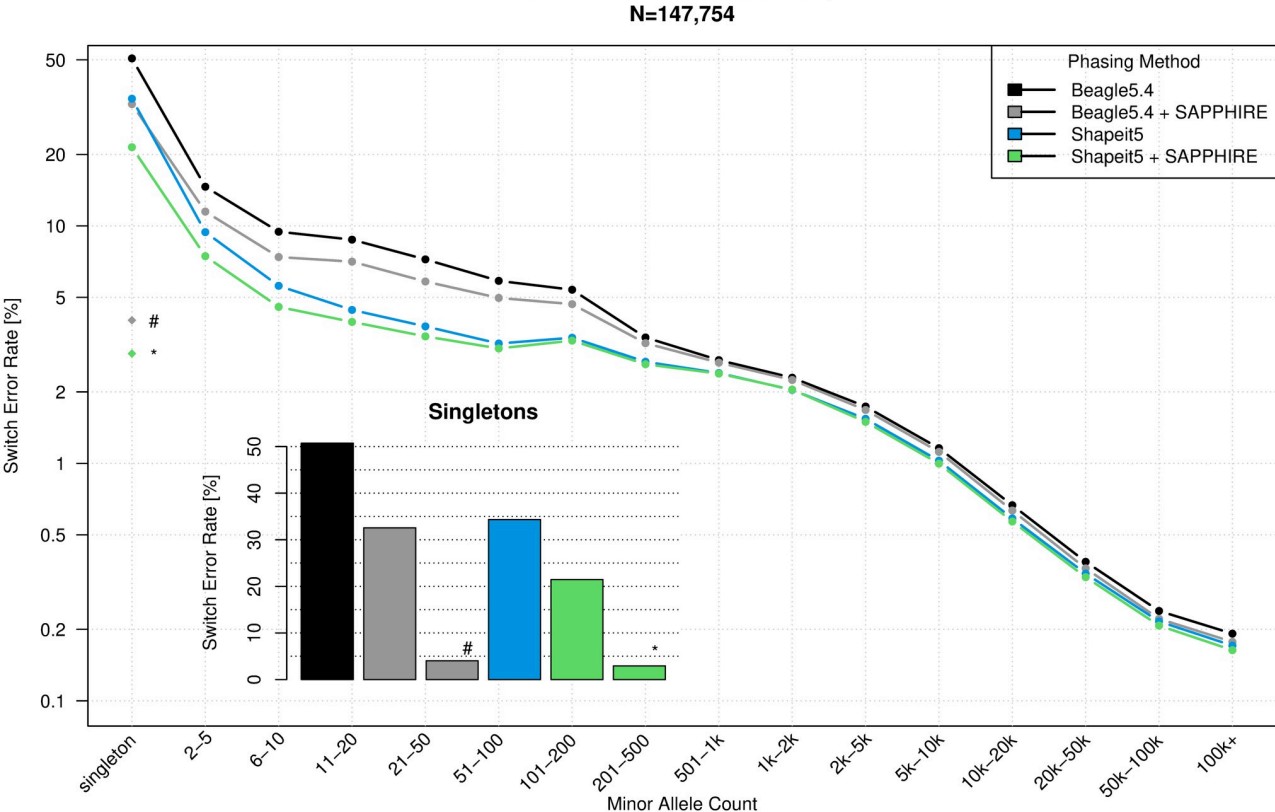

**Fig 2. Phasing performance.** Switch error rate (SER, y-axis, log-scale) of the polished haplotypes with SAPPHIRE (applied on SHAPEIT5 phased data in green, and on Beagle5.4 phased data in gray) compared to SHAPEIT5 (blue) and Beagle5.4 (black) stratified by minor allele count (x-axis) for the UK Biobank whole-genome sequencing chromosome 20 data. The * shows the switch error rate (2.9%) for SHAPEIT5 phased singletons rephased by the SAPPHIRE method (43% of all singletons), and the # shows the switch error rate (4.0%) for Beagle5.4 phased singletons rephased by the SAPPHIRE method (37.8% of all singletons).

approach, which comprises 43% of all singletons on the SHAPEIT5 haplotypes, the error rate drops to 2.9% (Fig 2). The same is true for the singletons phased by the Beagle5.4 method, the ones rephased by SAPPHIRE (37.8% of all singletons) have a SER of 4.0%. Overall, the results demonstrate that our method substantially improves the phasing quality at variants with a minor allele count below 200 (minor allele frequency (MAF) < 0.27%), with the highest boost observed at singletons. For the SHAPEIT5 haplotypes the phasing probability (PP) score given by SHAPEIT5 was used to define which variants were rephased. All genotypes phased with score below 0.99 (99% confidence) were rephased against the common (MAF > 0.1%) and high confidently phased rare variants. On the Beagle haplotypes all genotypes with a MAF below 0.1% were rephased against the remaining variants. A more in-depth look into the effect of MAF threshold is discussed in the Methods section.

## Assessment of the phasing quality of de novo mutations

To biologically validate the haplotypes and to showcase a possible downstream usage, we looked at de novo mutations (DNMs). DNMs are mutations that are not present in the parental genomes. They can occur in an egg or sperm cell of a parent, in the fertilized egg after the egg and sperm unite, or in another type of cell division during embryo development [13]. Note

that only a minority of DNMs are postzygotic as shown in a recent study assessing 6,034 DNMs in 91 monozygotic twins in which 97.1% of the DNMs were found in both twins [14] (and therefore are not postzygotic). DNMs are true singletons even in trio data sets as they are new mutations and do not appear in the parental genomes. However, these DNMs still require to be phased to either the paternal or maternal haplotype, which is impossible to achieve using parental genomes. All non-postzygotic (most DNMs) have a parental origin that can be deduced from the haplotype on which they lie. Multiple studies have shown that DNMs originate more often from the paternal side than from the maternal side, with a ratio of between $3:1$ and $4:1$ [14–17]. In our work, we evaluated the phase of 3,744 SNV singletons (de novo mutations) in the offspring of 93 trios from the UK Biobank 200k whole-genome sequencing data. Out of the 3,744 DNMs, 1,686 (45%) could be linked to a paternal or maternal haplotype using sequencing reads. Fig 3 summarizes the phase identification of DNMs. The phased DNMs show a total mean $1,271:415 \approx 3.06:1$ paternal/maternal ratio and a median ratio of $14:4 = 3.5:1$ paternal/maternal DNMs which is in line with previous findings. If we use the phase given to these 1,686 DNMs by SHAPEIT5 the total mean ratio is $1096:590 \approx 1.86:1$ and median ratio of $12:6 = 2$ paternal/maternal DNMs which is further away from the expected result because of phasing errors. It has also been shown that the number of DNMs in offspring is positively correlated with increasing paternal age at the time of conception [14–17]. The data from the 93 trios was used to assess this relation. Due to the age distribution within the UK Biobank (40–69 years), the age range of parents in trios is limited to young adults. However, results shown in Fig 4 still exhibit a positive trend with the age of the father at conception, although not significant ($\beta = 0.247$, $r = 0.153$, $p$-value $= 0.142$ and $\beta = 0.160$, $r = 0.105$, $p$-value $= 0.318$, before and after polishing). Without SAPPHIRE the phased data displays a spurious negative trend relative to the maternal age at conception ($\beta = -0.090$, $r = -0.071$, $p$-value $= 0.500$). After polishing, the negative trend relative to the age of the mother disappears ($\beta = -0.008$, $r = -0.009$, $p$-value $= 0.926$). These results conform with the observations in other studies [14–17]. The replication of these patterns demonstrates that the phasing of the SAPPHIRE method at singletons, variants at which statistical phasing performs poorly, is of high quality and can be used to reveal important biological properties.

## Number of rare variants that can be rephased through SAPPHIRE

Rephasing a rare variant requires close enough proximity to a common variant in order for both to be covered by the same sequencing reads or read pairs, as shown in Fig 1B. We therefore evaluated rare variants on chromosome 21 for 200,031 samples to see how many could be rephased using sequencing reads. We extracted a total of 11,471,867 heterozygous variants with a SHAPEIT5 confidence score below 0.99 (considered here as poorly phased with statistical phasing). Fig 5 shows the distribution of variants. Only SNVs (89.4%) were considered. In theory, other types of variants (e.g., short indels) can also be rephased but special care to the realignment model is needed to accommodate such cases. In total, 45.2% of the SNVs shared at least one high quality sequencing read or read-pair with a nearby common and accurately phased variant. These numbers are dependent on the sequencing technology and depth. In the UK Biobank participants were sequenced with an average coverage of 32.5× (at least 23.5× per individual) using Illumina NovaSeq sequencing machines [10]. Long read sequencing [18] could improve this even further but was not available for the UK Biobank. SAPPHIRE can be applied on low coverage sequencing data as well. As long as there is a sequencing read linking a low confidence genotype with a high confidence phased genotype SAPPHIRE can rephase it. In general, higher coverage will lead to better results. Technically, exome sequencing could be used with SAPPHIRE. However, as whole-exome sequencing covers only about 2% of the

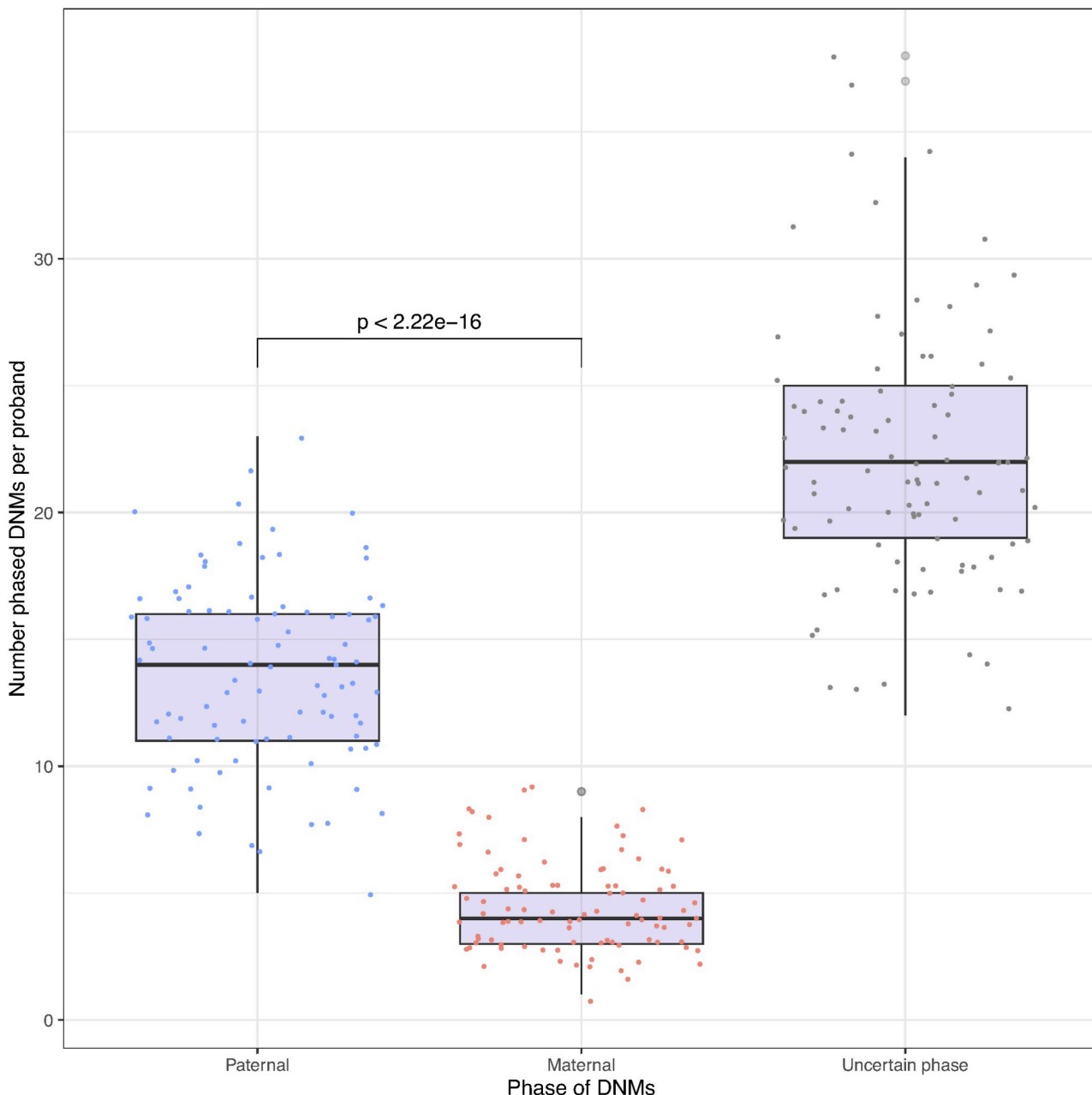

**Fig 3. Distribution of de novo mutations (DNMs) either on the paternal or maternal haplotype confirmed by WGS reads.** DNMs or uncertain phase are DNMs that have been phased with low confidence and have no WGS reads to validate their phase. The paternal/maternal median ratio is 14 : 4 = 3.5 : 1 paternal/maternal DNMs. *p*-value between paternal and maternal categories correspond to Wilcoxon test *p*-value.

genome and 80% of exons being shorter than 200 base-pairs [19] resulting in very short read fragments, the overall improvements in phase accuracy would be marginal. Out of the SNVs, 69.7% were phased correctly by SHAPEIT5, while 30.3% required the original phase to be corrected. Finally, Figs 6 and 7 show the distribution of phase corrections per minor allele count (MAC) bin, as counts and percentages, respectively. As expected, the singletons required more correction, as these are not possible to phase reliably through statistical models.

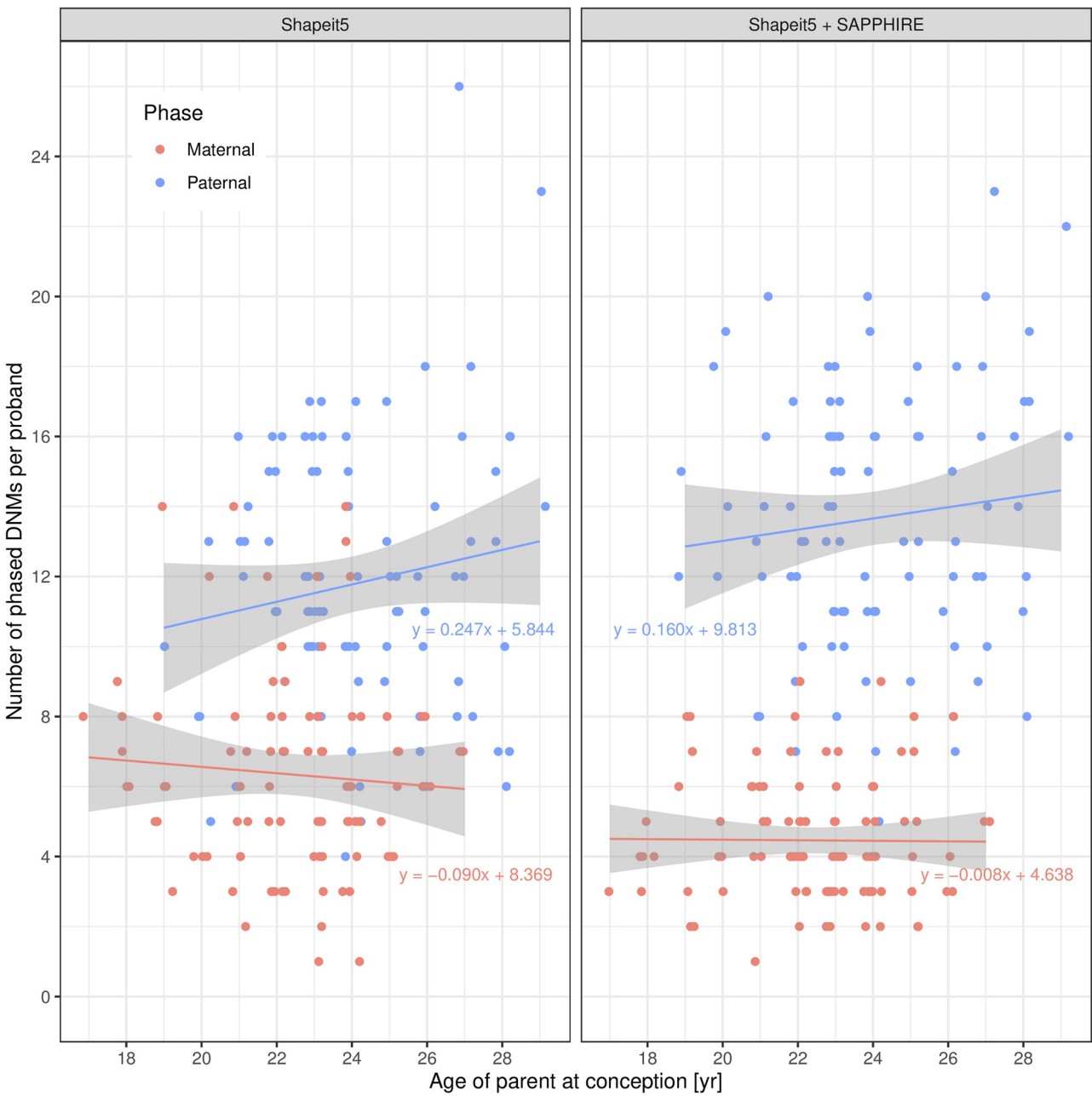

**Fig 4. Number of de novo mutations (DNMs) per proband grouped by parent of origin relative to the age of parent at conception.** A positive trend is observed relative to the age of the father at conception both before and after polishing. A spurious negative trend is observed relative to the age of the mother on uncorrected SHAPEIT5 data. This artifact disappears after phase polishing with SAPPHIRE. $N = 93$, DNMs = 1, 686. (A random ±0.25-year jitter is applied on the age for visualization only to avoid data points being merged into a single point, linear regressions are computed on real age which is provided at single year resolution).

## Cost of the SAPPHIRE method

All computations have been performed on the Research Analysis Platform (RAP) of the UK Biobank. The cost to phase 147,754 samples on chromosome 20 with SHAPEIT5 was £74.9 [3]. In contrast, phase polishing the resulting haplotypes with SAPPHIRE cost only £5.0

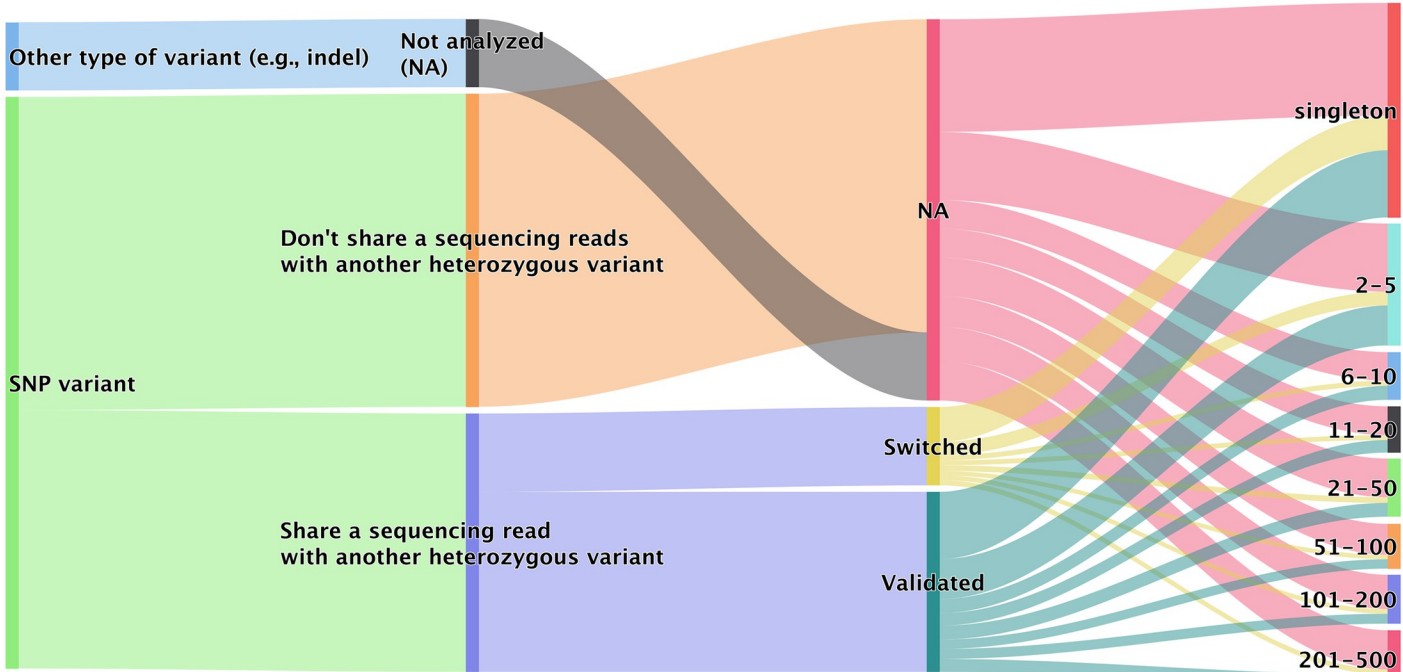

**Fig 5. Distribution of heterozygous genotypes phased by SHAPEIT5 with phasing probability (PP) score less than 0.99.** For the 200,031 sample release of the UK Biobank whole-genome sequencing calls, 11,471,867 heterozygous genotypes on chromosome 21 were extracted based on the phasing probability (PP) score given by SHAPEIT5. Out of all the genotypes with a PP score less than 0.99 only SNVs (89.4%) were considered for analysis, out of all SNVs 45.2% share at least one sequencing read or read-pair with a confidently phased genotype. Of all analyzed SNVs, 30.3% were phased incorrectly with regards to the other genotype on the reads. The last column displays the ratios of genotypes that were switched, validated, or not analyzed per minor allele count bin.

which is less than 10% of the original cost. Phase polishing the full set of 200,031 samples of the UK Biobank across all chromosomes requires going through 200,031 CRAM files (on average 18 GB per sample), representing over 3.5 Petabytes (3,669,126 Gigabytes) of data in total. The total cost for this step was £570 (details given in the methods section). We also estimated the cost of running WhatsHap [5] to extract phase sets on the same data set, and this would add up to at least £50,000 (see Methods). This demonstrates the need for a targeted approach compared to first extracting phase sets for all heterozygous variants from sequencing reads.

## Assessment of SHAPEIT5 phasing confidence

The confidence score given by SHAPEIT5 is a probability that ranges between 0.5 and 1.0. A score of 0.5 means full uncertainty in the phasing call (equivalent to a coin toss) while 1.0 means that the software is pretty certain on the call it reports. The SAPPHIRE method allows us to verify, based on sequencing reads, many of the original phase calls and we can collect statistics about their correctness. In practice, we binned the SHAPEIT5 phase calls by confidence scores (PP field) and computed the number of calls that were validated or invalidated by sequencing reads. We analyzed chromosome 21 for 200,031 samples and collected statistics for 4,634,625 phase calls. Fig 8 reports the percentages of phase calls that we validated or corrected using sequencing reads. Higher confidence scores come with lower error rates, ranging from 33.7% to 12.5%. For a phase confidence of 0.5, SHAPEIT5 performs better than a coin toss

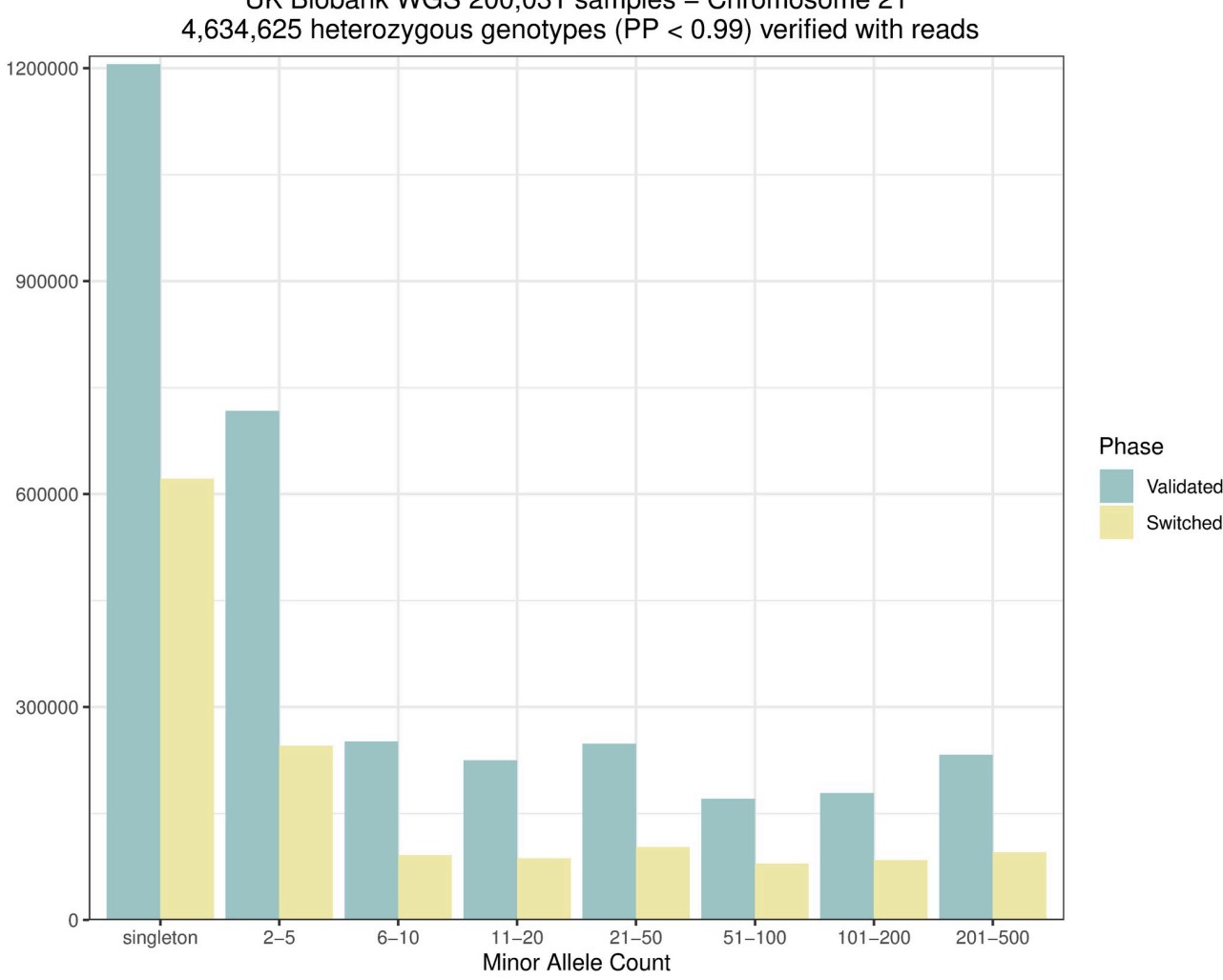

**Fig 6. Phase validation.** Number of heterozygous genotypes with the phase validated or switched grouped by minor allele count in the 200,031 samples data set on chromosome 21 when applying SAPPHIRE after SHAPEIT5.

with only one third of the calls being incorrect. For a phase confidence score between 0.9 and 0.99, SHAPEIT5 seems a little overconfident. As our validation set only covers 45.2% of all SNVs we analyzed the PP distribution within the other variant classes (variants not linked by sequencing reads and non SNVs) to make sure there was no bias towards a particular SHAPEIT5 PP-score. The distributions are similar in all three classes as reported in supporting S1 Fig. These results show that the SAPPHIRE method can also be used to evaluate and re-calibrate phasing confidence scores models.

## Discussion

We present SAPPHIRE a novel approach to improve the phasing accuracy at rare variants within population-scale data sets initially phased with statistical methods. We show that we can reduce switch error rates for variants with low-occurrence frequencies with the most notable impact at extremely rare variants and singletons. Our method also delimits the subset of

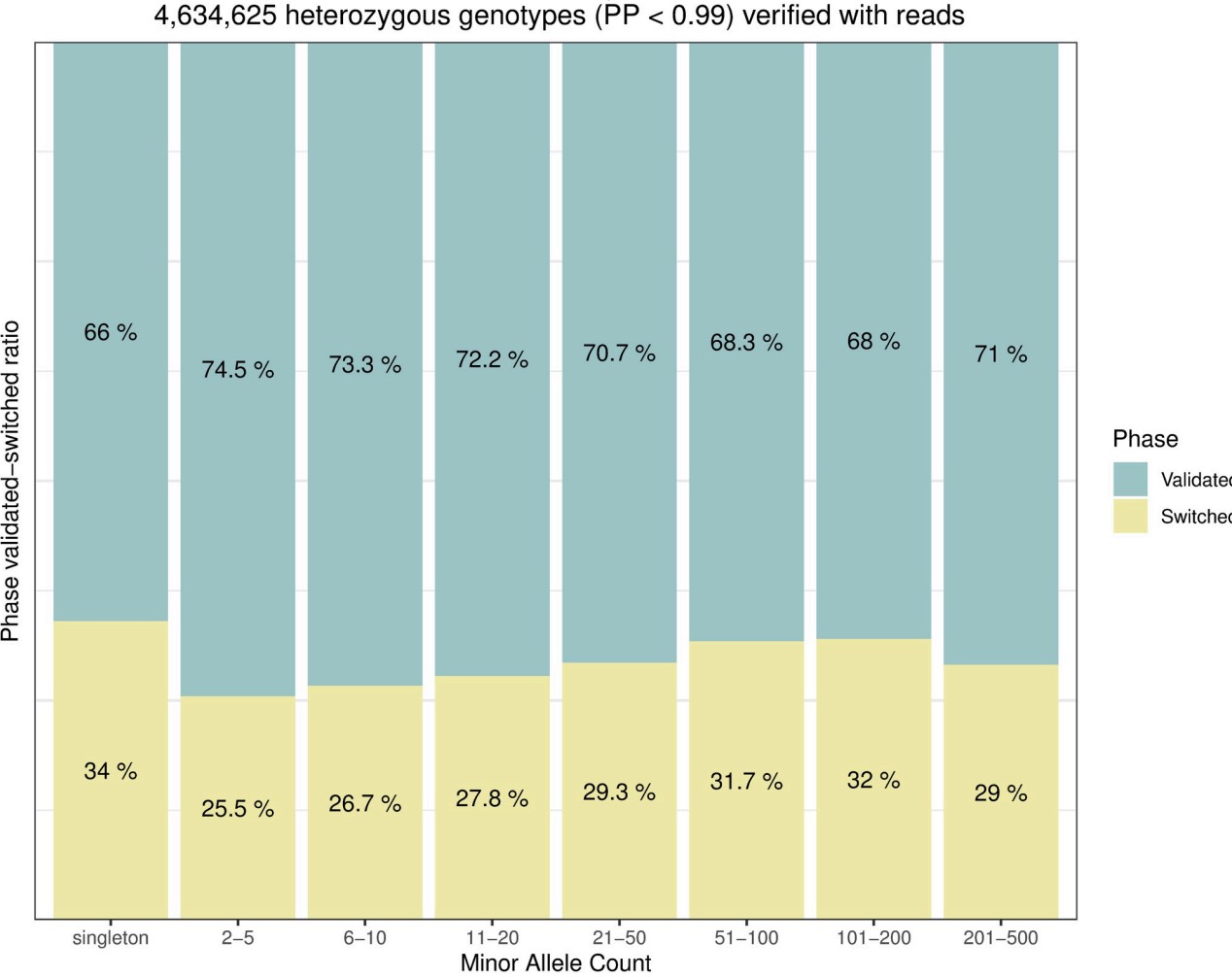

**Fig 7. Phase validation distribution.** Distribution of validated/switched phase calls of 4,634,625 genotypes grouped by minor allele count in the 200,031 samples data set chromosome 21 when applying SAPPHIRE after SHAPEIT5.

phased alleles that have been validated by sequencing reads, which makes it possible to discern them from statistically phased variants. We show that our method is scalable and can be used on Petabytes of sequencing data thanks to a targeted approach where only alleles of low confidence are analyzed. Phase polishing of the UKB WGS data set with 200,031 samples phased with SHAPEIT5 through SAPPHIRE was achieved with a compute cost of £570, or less than £0.003 per sample. The rephased data set shows reduced error rates on all rare variant frequency bins with major improvements on the phasing of singletons. SAPPHIRE not only improves overall estimated haplotype quality but can also be used as a benchmark for phasing in general as it can report the number of discordances between estimated haplotypes and phase relationships observed within sequencing reads.

The subset of read validated phased rare variants allows for new downstream analyses which were not possible with the lower accuracy of statistical methods, especially for singletons. To demonstrate the impact of accurately phasing singletons we replicated two key observations of de novo mutations: (i) we observed the expected paternal/maternal imbalance in the

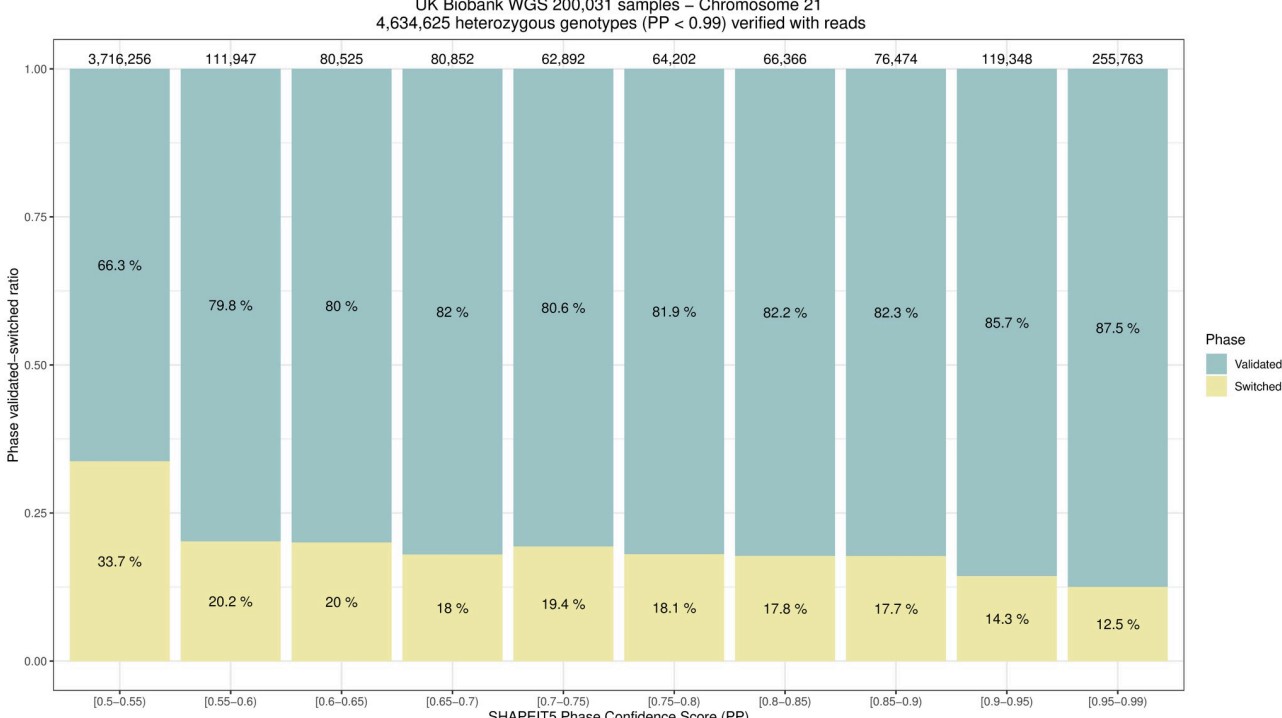

**Fig 8. Assessment of SHAPEIT5 phasing confidence score (<0.99) with sequencing reads.** We evaluated the phasing confidence scores smaller than 99% given by SHAPEIT5 on its phase calls with sequencing reads. Out of 11,471,867 heterozygous genotypes on chromosome 21 for 200,031 samples we were able to check the phase of 4,634,625 with sequencing reads. The ratio of validated/switched phase calls is reported per phase confidence score bins as given by SHAPEIT5.

origin of the de novo mutations, and (ii) we found a positive correlation with the father's age at conception and the number of de novo mutation in the proband. These observations could not be replicated through statistical phasing alone. We predict that our method will allow for further analyses of rare variation such as discovery of compound heterozygous gene knockouts.

We show that by combining statistical methods and targeted read based methods, we can provide highly accurate haplotype estimates at a low cost. The method can be applied on a per sample basis, on a sample subset of interest or on a whole population. If long read sequencing data becomes available for the samples, the method would allow linking even more variants and further improve the estimated haplotypes.

We believe that phase polishing large statistically phased data sets is the most effective method to date. By combining the effectiveness of statistical methods for common variants and the precision of reads for rare variants we achieve the highest possible accuracy. The order of the methods is of crucial importance, because of the high cost of processing large amounts of sequencing data, it is best to only target sites of low confidence given by the statistical phasing method (or of low minor allele frequency by proxy) in order to avoid spending time on sequencing data where it unnecessary. We estimated that the SAPPHIRE method has a cost at least a hundred times lower than current read-based methods and have shown that it is possible to polish all autosomes for 200,031 samples for a total of £570 improving the overall phasing quality of the data set especially at rare variants and even singletons.

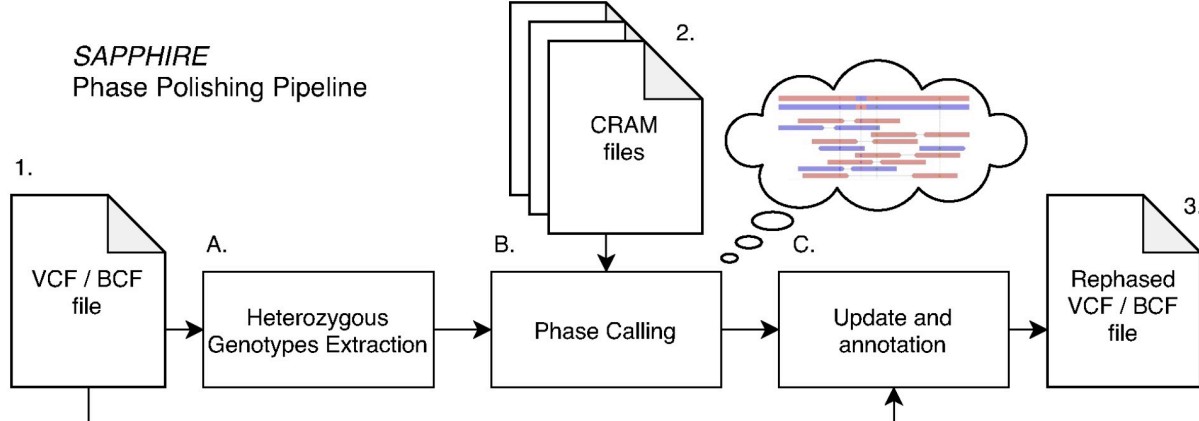

**Fig 9. SAPPHIRE phase polishing pipeline.** From an input VCF/BCF file **1** (e.g., whole chromosome) heterozygous genotypes are extracted **A** to be rephased alongside their closest neighboring heterozygous genotypes. The phase calling stage **B** accesses the whole-genome sequencing files (CRAM files **2**) to call the phase of extracted heterozygous genotypes in relation to their neighbors. Finally, the original VCF/BCF file 1 is updated and annotated **C** to generate the final rephased VCF/BCF file **3**.

## Methods

Phase polishing through the SAPPHIRE method is as follows: First, heterozygous genotypes from all samples are extracted. Then, the phase for each pair of genotypes with overlapping sequencing reads is verified, with common variants as reference for phase verification. Rare variants are checked against common ones. If sequencing reads clearly show a reversed phase, SAPPHIRE corrects it and the read count supporting the phase is reported. Unchanged variants also have their read count reported for phase call confidence. Finally, all switched or validated genotypes are updated in the original VCF file to include the new updated phase and supporting read count. The pipeline is depicted in Fig 9. The pipeline steps are discussed in the methods below. The extraction algorithm (step A.) is described in detail in supporting information file S1 Algorithm. *Heterozygous variant extraction from VCF/BCF file*. The phase calling algorithm (step B.) is described in detail in supporting information file S2 Algorithm. *Phase calling (polishing)*. Update and annotation is straightforward and described below.

### Extraction of heterozygous genotype data for polishing

First, for each sample, all heterozygous genotypes phased by SHAPEIT5 with a phasing confidence score (PP field in VCF) below 0.99 (99% confidence) were extracted alongside their closest neighboring heterozygous genotypes (up to two before and two after, the closest neighboring heterozygous variants are different for each sample) regardless of their phasing confidence score. The exact algorithm is described in supporting information file S1 Algorithm. The extracted heterozygous genotypes and their neighbors are stored in a binary format and the variant loci information is kept in BCF format (without any sample). Splitting the information this way allows for faster random access [20]. The generated files are then passed along to the next stage, the phase caller, which will query whole-genome sequencing reads to verify the phase calls.

### Phase calling and polishing of extracted sites

Extracted heterozygous sites are filtered on variant type and only single nucleotide variants (SNVs) are processed, as shown in Fig 5. For each site a pileup of sequencing reads is done

through HTSLIB [21] similarly to the pileup function of BCFTools [22]. For each site that has a phasing confidence score below 0.99, evidence from sequencing reads is queried and the number of reads that confirm or invalidate the phase of the site is counted. If there is enough evidence (at least one high quality mapped sequencing read and no contradicting reads) to show a phase error, the phase is flipped as shown in Fig 1B. Evidence, the number of reads that confirm the new phase, is recorded. For all sites that are phased correctly, the number of reads is also recorded in the phase confidence score to show that the phase has been validated by sequencing reads. The exact algorithm is described in supporting information file S2 Algorithm.

### Effects of errors in original phasing and ambiguities

In the case where a common variant is erroneously phased from the start, a rare variant sharing sequencing reads will be rephased on the wrong haplotype, introducing a flip error. However, this does not change the overall number of switch errors and will correct the phase relationship between the two nearby variants. As to which is better depends on downstream analysis. This depends on if the phase relationship between close variants is more important or the overall phase of the rare variant. For example, in studies on compound heterozygous mutations, local phase relationship is more important. In cases where multiple common variants are close to a rare variant and one of them is erroneously phased, the phase calling algorithm will rephase the rare variant according to the biggest number of sequencing reads that support a given phase.

### Effect of MAF threshold on phase quality

SHAPEIT5 provides a phasing confidence (PP) score for rare variants (MAF below 0.001) which is leveraged by SAPPHIRE to decide which one to rephase based on close common and high confidence variants (PP $>= 0.99$). However, other phasing methods do not provide this score. To allow SAPPHIRE to be applied nonetheless we have to rely on another metric. For this we chose MAF as a proxy, i.e., rephase all variants below the MAF threshold in relation to the variants above the threshold. As genotypes at low MAF variants exhibit higher error rates, we chose MAF 0.001 as the default threshold. This ensures that the variants that are used as the reference for rephasing have a low mean SER (less than 5% for all methods as shown in Figs 2 and 10. Note: 0.001 MAF corresponds to 295 MAC). To validate this choice, we have evaluated SAPPHIRE on both the Beagle and SHAPEIT5 phased haplotypes with a 5× geometric progression for the MAF threshold i.e., 0.0002, 0.001, and 0.005 (MAC 59, 295, 1478 respectively). The results are shown in Fig 10. We found that a 0.001 MAF threshold gave the best results, going for lower MAF results in several poorly phased genotypes left untouched, resulting in higher SER in the 51–200 MAC bins. A MAF threshold of 0.005 results in a very slightly worse overall SER. The very close results between 0.001 and 0.005 show that the method is robust and gives good results even when a sub-optimum threshold is chosen. Overall, the phase confidence (PP) score given by SHAPEIT5 results in the lowest SER across all bins. This score also allows SAPPHIRE to use confidently phased rare variants as reference for the phase polishing and results in a larger number of variants being rephased, as can be seen for singletons where 43% of all could be linked to a confidently phased variant, as opposed to 37–38% for the MAF approach.

### SAPPHIRE performance in complex genomic regions

To evaluate the SAPPHIRE method in complex genomic regions we extracted several metrics of SAPPHIRE on chromosome 6. We compared the results within the human leukocyte

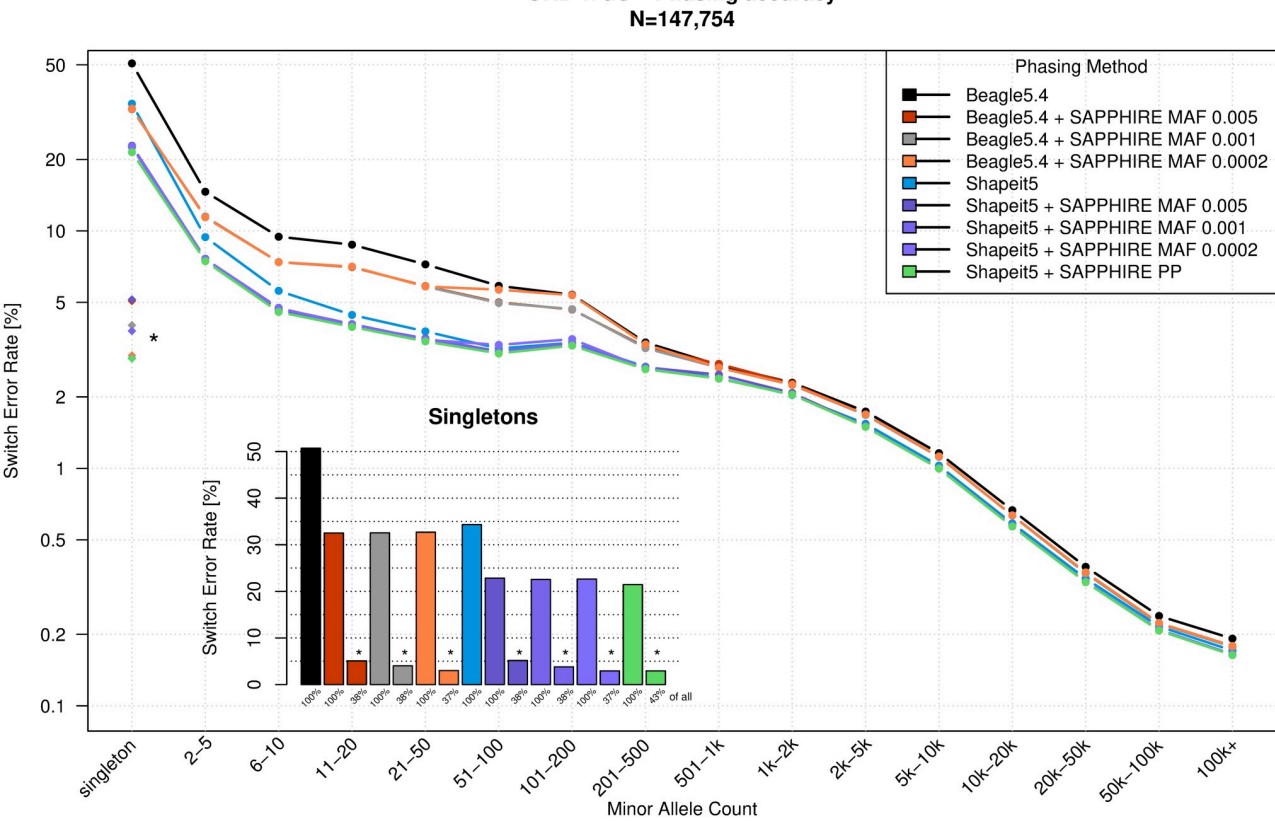

**Fig 10. Phasing performance.** Switch error rate (SER, y-axis, log-scale) of the polished haplotypes with SAPPHIRE (applied on SHAPEIT5 and on Beagle5.4 phased haplotypes) with different MAF thresholds, stratified by minor allele count (x-axis), for the UK Biobank whole-genome sequencing chromosome 20 data. The * shows the switch error rate for singletons rephased by the SAPPHIRE method (37–43% of all singletons).

antigen (HLA) region, which exhibits a high level of polymorphisms and mutations [23], to the rest of the chromosome. Supporting file S2 Fig shows the low-PP (<0.99) genotype density over 100kB regions, where we observe a uniform distribution with a spike around the centromere. Of the low-PP genotypes the percentage that could be linked to a high-confidence of common heterozygous genotype is 43.7% which is coherent with our findings on chromosome 21 (45.2%) in Fig 5. This number increases within the HLA region because as this region has more mutations than the rest of the chromosome, there will also be more common and confidently phased variants. This results in the biggest percentage of low-PP genotypes being linked on the whole chromosome, allowing SAPPHIRE to rephase more genotypes, up to 71.3% (per 100kB region). Of all the genotypes, 28.5% had their phase switched by SAPPHIRE, which again is coherent with the results on chromosome 21 (30.3%). The average number of switched genotypes within the HLA region is similar to the rest of the chromosome. The average PP-score given by SHAPEIT5 within the HLA region is higher than in other regions, which is explained by the higher number of overall variants in the HLA landscape. This allows for better statistical phasing and therefore higher confidence scores. This shows that the SAPPHIRE method actually benefits from regions with higher numbers of mutations as this means more opportunities to link close heterozygous genotypes through sequencing reads, especially for SNVs. If SAPPHIRE would be extended to handle complex mutations such as structural variations (SVs), complex regions would need to be reevaluated but for SNVs regions with higher

number of mutations allow to phase polish more genotypes with overall results similar to the rest of the chromosome.

### Read quality filtering

In order to avoid lowering the quality of the input data sets to be polished, we applied a very strict filter on which sequencing reads are allowed to be used as evidence. A sequencing read-pair would be rejected by the SAPPHIRE algorithm if: Mapping quality score (MAPQ) is below 50, or one of the sequencing reads in the pair is unmapped, or a read is unpaired, or the read pair has incorrect orientation (e.g., both reads on the forward strand), or a read has the duplicated flag set. This way only very high quality aligned reads remain, which at one side results in less genotypes getting their phase validated or switched, but on the other side, newly validated or switched genotypes are of very high confidence. Adjustment of this read quality filtering could allow it to phase polish more genotypes, evaluation of a more relaxed filtering is left for future works.

### Scatter-Gather parallelism

In order to reduce the time needed to process all samples on the UK Biobank RAP, the extracted data is split in batches of samples, e.g., 1,000 samples, 5,000 samples., 10,000 samples. Because all samples are processed individually with their respective whole-genome sequencing data, this process can be multi-threaded up to a certain degree dictated by the access band-width to the whole-genome sequencing files per node. For example, we empirically found out that running a dozen threads on a four CPU node would give the most efficient node usage. The number of threads being higher than the CPU count is so that enough threads are working while other threads are waiting for data to arrive over the network.

### Accessing a large number of sequencing files

Since the whole-genome sequencing read files (in CRAM format [24, 25]) are on average more than 17 GB per sample resulting in 3,669,126 GB for all 200,031 samples, naive approaches are not tractable. The jobs running SAPPHIRE were split per chromosome in batches of 1,000, 5,000, 10,000 samples based on chromosome size (details in S1 Table). Each job therefore requires access to at least 1,000 CRAM files. Jobs on UK Biobank RAP are run in compute nodes (backed by Amazon AWS instances) and can access CRAM files in two ways, first, by downloading the file, then processing it, second, by mounting the UK Biobank data as a read-only network file system. The first approach is not viable as it would first require a large amount of storage on the compute node and would spend a long time moving data before processing (1,000 samples ≈17 TB of data). Therefore, CRAM files were accessed over a network file system. HTSLIB, which allows for random access in indexed CRAM files and this enables the processing of a large number of samples on a single node. The main bottleneck becomes network speed, measured at about 50 MB/s on average. Thanks to the random access features of the CRAM format and HTSLIB, only the required parts of the CRAM files are accessed. The CRAM format also provides a checksum for each of its internal data records, allowing it to detect errors in network transfers and act accordingly.

### Update and annotation of the original VCF/BCF file

Finally, the original VCF/BCF file is updated with the new phase and phase confidence scores. This stage also checks that no errors have been introduced (e.g., a heterozygous genotype call becoming a homozygous call or that a high confidence heterozygous site would have been

rephased). The new phase confidence score is defined as follows: old confidence score plus number of reads that confirm the phase plus one. So, all genotypes with a score over 1.0 are polished genotypes. The original phase confidence can be extracted by looking at the fractional part, with the edge case of 1.0 which is a phase confidence of 1.0, which we don't rephase.

## Data

We used the whole-genome sequencing data available on the UK Biobank RAP as pVCF files that were called with GraphTyper [26]. We used two different releases of the UK Biobank WGS data: (a) the first release of 150,119 individuals, to compare the SAPPHIRE method to the original phasing with the SHAPEIT5 method [3], and (b) the newest release of 200,031 individuals. For the release of the 150,119 individuals with WGS data available, we performed quality control of this data as described previously [3, 27]. Briefly, we (i) used BCFtools [22] to split multi-allelic variants into bi-allelic variants, (ii) removed SNPs and indels having more than 10 percent of missing genotypes, Hardy-Weinberg p-value $< 10^{-30}$ and an excess of heterozygosity less than 0.5 or greater than 1.5. Additionally, we (iii) filtered out variants with AAscore $< 0.5$, and (iv) kept only variants with the PASS FILTER tag. In total, this filtering retained 603,925,301 variant sites across 150,119 individuals. For our analysis, we kept only individuals being also genotyped with the UK Biobank Axiom array. Additionally, since we used white British trios and duos ($N = 31$ and 432 individuals, respectively) to assess the accuracy of the method as previously described [3], we removed parental genomes from the data. This resulted in a total of 147,754 individuals. The quality control of the second UK Biobank WGS data release (i.e., 200,031 individuals) has been performed as part of the official release of the phased data by the authors of SHAPEIT5 [3]. Although very similar to the previous quality control, it has been optimized for phasing accuracy and only variants with AAscore $> 0.8$ were kept. In this release, family phasing was applied in order to use parental genomes to phase the offspring ($N = 93$ trios and $N = 915$ duos).

## Validation of haplotype estimates

To validate the haplotype estimates and compare the accuracy of methods, we used parent-offspring trios and duos that we identified using the kinship estimate and the IBS0 provided in the UK Biobank Axiom array release. Specifically, we infer parent-offspring relationships for any pair of individuals having kinship coefficient lower than 0.3553 and greater than 0.1767, and IBS0 lower than 0.0012 [1, 3, 28]. We then used the age to infer the direction of the relation-ship [29]. Additionally, we kept only white British individuals for which the ancestry was confirmed by Principal Component Analysis (PCA). It resulted in 31 trios and 432 duos across the 150,019 individuals, and 93 trios and 915 duos across the 200,031 individuals. The validation process consisted in phasing the data set excluding parental genomes, and using parental genomes to assess how close are the estimate from the true parental haplotypes, measured as a switch error rate (SER) which is defined as the fraction of successive pairs of heterozygous genotypes being correctly phased [3]. Alternatively, we used genotype imputation to assess the accuracy of our method regardless of family information. Notably, this allows us to validate our haplotypes for the second data release in which parental genomes were kept. For this purpose, we selected 1,000 individuals of white British ancestry that are unrelated to any other sample in the data set and for which SNP array data is available, as traditionally done in imputation experiments [27]. We removed these individuals from the phased data sets, constituting reference panels of 146,754 and 199,031 individuals. We used the UK Biobank Axiom array data as input for genotype imputation. We quality control the input data using the UKB SNPs and samples QC file (UKB Resource 531) as

previously described [3]. We filtered out (i) variants that were not kept in the official phasing of the Axiom array data [1] and (ii) variants with a difference greater than 0.1 in allele frequency between the Axiom array data and the reference panel [3]. We used IMPUTE5 [30] for genotype imputation of the SNP array data and the concordance tool of GLIMPSE2 [27] to assess imputation accuracy.

## Compute nodes and cost

The heterozygous genotype extraction (A in Fig 9) can be run on a single node per chromosome. For all samples the compute node was chosen on storage size due to the size of the input VCF files. The phase calling and polishing (B in Fig 9) is done per batch of 1,000–10,000 samples per node depending on chromosome size. Finally the update of each VCF file is run on a single node per chromosome (C in Fig 9). The total compute time and compute cost (time × price per node) is reported in S1 Table. In order to polish the phase of all 200,031 samples in the UK Biobank the total cost was £570 with ≈ 60% of the cost spent on the actual phase calling with WGS data in the SAPPHIRE method (B in Fig 9). Compute nodes for stage A were chosen on lowest cost nodes that would provide enough solid-state storage to store the input VCF file. Compute nodes for stage B were selected empirically to provide the best price-performance with relation to the network speed of the network attached storage that holds all the WGS data. Finally for stage C the nodes were chosen based on the storage for both input and output files. Note: There are two priorities *on-spot* (low priority) and *on-demand* (high priority) for nodes on the UK Biobank RAP and the difference in cost is about five-fold. Low priority jobs can be interrupted if there are no nodes left for high priority jobs. If this happens, the interrupted job will be relaunched with high priority, resulting in higher cost. This explains the variability of costs reported in S1 Table. The final cost of running the SAPPHIRE method pipeline per sample on the 200k UK Biobank release was £570 / 200,031 ≈ £0.00285, less than a single penny per sample.

## Cost estimation of read based phasing

Alternatively, to phase polishing statistically phased haplotypes it would be possible to apply read based phasing first, e.g., with WhatsHap [5] to generate phase sets and then in a second pass apply a statistical method that takes into account phase sets, e.g., SHAPEIT4 [6]. Running WhatsHap for a single sample on chromosome 22 with whole-genome sequencing took 14min43 on a UKB RAP `mem3_ssd2_v2_x8` node. WhatsHap reported: 157s reading CRAM, 85s parsing VCF, 25s selecting reads, 50s phasing, 90s writing VCF, and 476s uncategorized. The maximum memory usage was 10.8 GB. This benchmark was run with a VCF already sub-sampled to a single sample because opening the 200,031 sample VCF file with WhatsHap to phase a single sample did run out of memory on the 64 GB node. Note that extracting a single sample from the 200,031 sample VCF file took 262 minutes with BCFTools on a UKB RAP `mem2_ssd2_v2_x2` node with files stored locally on the SSD. We can estimate running WhatsHap alone for 200,031 samples will take about 50,000 CPU hours (200, 031 × 15min). Given the memory requirements per sample we can run up to five processes in parallel on a single 64 GB node reducing the effective time to 10,000 hours. We can compare this to the SAPPHIRE method runtimes in S1 Table. It takes 9h19min to extract the positions to phase polish for all samples on chromosome 22. The phase polishing itself takes 7h14 on 21 nodes, so 153 hours, followed by a final update of the VCF with all samples, which takes 24h28min. Therefore, the total time is 187 hours to phase polish 200,031 samples from VCF to VCF. If we compare the 10,000 hours running WhatsHap to 187 hours of phase polishing we can say that running WhatsHap would take at least 50× more time than SAPPHIRE and this

does not even include time to extract the samples from the VCF, nor applying the phase sets to another method, nor consolidating the results. The cost of SAPPHIRE for chromosome 22 was £4.47, the estimated cost for WhatsHap on chromosome 22 is £528 (10,000 hours at a £0.0528 per hour rate, the lowest rate for a 64 GB instance in the UKB RAP). The estimated cost is at least 100× more than SAPPHIRE. The SAPPHIRE method efficiency comes from several factors: First, it only considers variants with low phasing confidence (or low MAF as a proxy) where WhatsHap considers all heterozygous variants. Second, the extraction step of the SAPPHIRE method converts the input VCF to a sparse binary format that allows extremely rapid parsing of variants. Third, extraction is done directly as a stream to a file, therefore memory usage of the SAPPHIRE pipeline is very low and the extraction can be run on nodes with very limited memory (e.g., 8 GB) whereas WhatsHap failed to load a single sample from the 200,031 samples VCF on a 64 GB instance. Therefore, we can conclude that a lower bound for the cost of using a read based phasing method would be at least 100× more expensive than phase polishing with SAPPHIRE. Phase polishing all autosomes of 200,031 samples of the UKB WGS data set with SAPPHIRE did cost £570, an estimation for a full read based phasing method would be at least £57,000.

## Supporting information

**S1 Table. Cost of the SAPPHIRE Phase Polishing pipeline on the 200k UK Biobank release.** The cost for all three steps of the phase polishing pipeline for chromosomes 1–22 of the UK Biobank 200k release. The final number of polished genotypes (GTs) and rephased GTs is given for all chromosomes. Notes: Five-fold increases in cost are for jobs that were interrupted and had to be relaunched with higher priority. For example, extraction on chromosome 6 and 7 almost take the same time on the same machine but the job of chromosome 6 was interrupted and relaunched at a higher priority (and cost). Chromosome 3 had the phase calling jobs split into batches of 10,000 samples as a test, which showed that it was better to split big chromosomes in more batches of smaller sample size. This has two advantages, first, the wall clock time is reduced and second, the cost is reduced because with the large sample size, the jobs had to run for a long time and were interrupted and had to be relaunched with high priority which increased the cost.
(XLSX)

**S1 Algorithm. Heterozygous variant extraction from VCF/BCF file algorithm description and complexity analysis.**
(PDF)

**S2 Algorithm. Phase calling (polishing) algorithm description and complexity analysis.**
(PDF)

**S1 Fig. SHAPEIT5 phase confidence PP-score distributions on chromosome 21 for the 200,031 samples release of the UK Biobank.** Distributions for the three leftmost classes of variants in Fig 5. In linear scale (top) and logarithmic scale (bottom).
(PDF)

**S2 Fig. SAPPHIRE on chromosome 6 of the 200k UK Biobank data set.** Top panel: Density of low-PP per 100kB region over chromosome 6. Middle panel: percentage of low-PP variants that share a read-pair with a common variant (green) and percentage of variants switched by the SAPPHIRE method (purple). Bottom panel: Plot of PP value for 100,000 randomly sub-sampled genotypes out of 45,376,832 (black) and average PP score per 100kB region for all

genotypes (red). Plotted with karyoploteR [31].
(PDF)

**S1 Data. Script for figures 2 and 10.**
(ZIP)

**S2 Data. SHAPEIT5 benchmark data for figures 2 and 10.**
(ZIP)

**S3 Data. Beagle5.4 benchmark data for figures 2 and 10.**
(ZIP)

**S4 Data. SHAPEIT5 and SAPPHIRE PP benchmark data for figures 2 and 10.**
(ZIP)

**S5 Data. MAF 0.005 benchmark data for figures 2 and 10.**
(ZIP)

**S6 Data. MAF 0.001 benchmark data for figures 2 and 10.**
(ZIP)

**S7 Data. MAF 0.0002 benchmark data for figures 2 and 10.**
(ZIP)

**S8 Data. Script and data for figures 3 and 4.**
(ZIP)

**S9 Data. Script and data for figures 5, 6, 7, 8.**
(ZIP)

## Acknowledgments

The benchmarks on the UK Biobank data have been conducted using the UK Biobank resource under application number 66995.

## Author Contributions

**Conceptualization:** Rick Wertenbroek, Olivier Delaneau.

**Data curation:** Rick Wertenbroek, Robin J. Hofmeister, Olivier Delaneau.

**Formal analysis:** Olivier Delaneau.

**Funding acquisition:** Yann Thoma, Olivier Delaneau.

**Investigation:** Rick Wertenbroek, Olivier Delaneau.

**Methodology:** Rick Wertenbroek, Olivier Delaneau.

**Project administration:** Olivier Delaneau.

**Resources:** Yann Thoma, Olivier Delaneau.

**Software:** Rick Wertenbroek.

**Supervision:** Ioannis Xenarios, Yann Thoma, Olivier Delaneau.

**Validation:** Rick Wertenbroek, Robin J. Hofmeister, Ioannis Xenarios, Olivier Delaneau.

**Visualization:** Rick Wertenbroek.

**Writing – original draft:** Rick Wertenbroek, Robin J. Hofmeister, Ioannis Xenarios, Yann Thoma, Olivier Delaneau.

**Writing – review & editing:** Rick Wertenbroek.

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
