## [Decision Letter · Decision Letter 0]

20 Feb 2024

Dear Dr Wertenbroek,

Thank you very much for submitting your Research Article entitled 'Improving population scale statistical phasing with whole-genome sequencing data' to PLOS Genetics.

The manuscript was fully evaluated at the editorial level and by independent peer reviewers. The reviewers appreciated the attention to an important problem, but raised some substantial concerns about the current manuscript. Based on the reviews, we will not be able to accept this version of the manuscript, but we would be willing to review a much-revised version. We cannot, of course, promise publication at that time.

If you decide to revise the manuscript for further consideration at PLOS Genetics, please aim to resubmit within the next 60 days, unless it will take extra time to address the concerns of the reviewers, in which case we would appreciate an expected resubmission date by email to plosgenetics@plos.org.

We are sorry that we cannot be more positive about your manuscript at this stage. Please do not hesitate to contact us if you have any concerns or questions.

Yours sincerely,

Yun Li

Academic Editor

PLOS Genetics

Michael Epstein

Section Editor

PLOS Genetics

The manuscript is of potential broad interest. But the reviewers pointed out multiple major issues. Please carefully address the reviewers' comments.

Reviewer's Responses to Questions

**Comments to the Authors:**

Reviewer #1: In this manuscript, Wertenbroek et al. proposed an interesting re-phasing method, SAPPHIRE, to correct phasing errors specifically for rare variants, based on the phased output from SHAPEIT5. It takes sequencing read data from CRAM files in addition to the genotype data in VCF/BCF format, and borrows information of a nearby common variant to validate the phasing accuracy. The intrinsic idea of the method is straightforward with a clearly improved phasing accuracy, but the authors should provide more evidence for its broad utility.

Major:

1. I am wondering how the nearby common variants were selected. How the "common" was defined? Was the choice based on allele frequency? Please confirm, SHAPEIT5 does not provide phase accuracy score for common variants, right? SAPPHIRE assumes those common variants were accurately phased, though largely fine, some phasing error may still exist. What if it picks a common variant with phasing error?

2. SAPPHIRE was built based on the phasing accuracy score from SHAPEIT5 by extracting sites with accuracy score < 0.99 for all the rare variants. Though the authors mentioned MAF could be a proxy, they didn’t perform analyses based on MAF. I am wondering if the authors could demonstrate the advantages of SAPPHIRE using output from Beagle5 without accuracy score information, especially given that Figure 2 shows Beagle5 achieved the largest phasing error. Will SAPPHIRE similarly achieve improved accuracy?

3. According to Figure 5, SAPPHIRE could only work for 45.2% of the SNVs which share a sequencing read with another heterozygous variant. Did the authors assess if there are any differences between the two sets, in terms of both phasing accuracy score from SHAPEIT5 and true phasing accuracy from families? If significant difference exists, the section “assessment of SHAPEIT5 phasing confidence” will be biased, since only a non-randomly selected subset could be assessed.

4. Are there any variations across the genome regarding the phasing accuracy? Does SAPPHIRE also improve phasing accuracy for some highly complicated genome regions, e.g., the HLA region?

Minor:

1. Figure 2: since SAPPHIRE is an phasing enhancement method rather than a phasing method, I suggest the figure legends should change from SAPPHIRE to SHAPEIT5+SAPPHIRE.

2. In the section of DNMs, when assessing the relationship between number of DNMs in offspring and paternal/maternal age, the authors only reported results after SAPPHIRE correction. How about before SAPPHIRE correction using only SHAPEIT5 output?

3. The authors assessed the performance of SHAPEIT5 confidence score by different bins for variants with score < 0.99. Are the sites with score > 0.99 truly trustworthy? Especially given the last bin with the highest scores also show switch rate 12.5%. Again, this may be biased depending on whether there is significant differences between the two sets of variants (major point 3). Also, the last bin should be [0.95, 0.99) if the authors didn’t evaluate variants with scores > 0.99.

Reviewer #2: Wertenbroek et al proposed an innovative statistical phasing method based on whole-genome sequencing data. I think the method looks valid considering that they have compared their software with other software in literature, and applied to PB-level big data. However, the methodology description is unclear such that I have a lot of questions after reading the manuscript.

Major Comments:

1. SAPPHIRE is the author's proposed software. However, I did not find the description of statistical methods used in the software. What is the innovation and advantage of SAPPHIRE? I only find they make corrections on phasing error, but do not know whether this correction improves statistical phasing, or just via post simple adjustment. Please specify in detail the methods used in the software.

2. It is important for the authors to report computational time for different computational nodes. But I did not have full picture of computational time. Can the authors specify what growth pattern of computational complexity with respect to sample size. The current version did not provide any sentence on it.

3. I think SAPPHIRE may do post-phasing adjustment, or how they make use of sequencing data? Details are needed.

4. The real data part is ok. I know it is a big data, and still there is lack of validity.

5. What is the evaluation criteria in comparing multiple software, though Fig 5 and 6 seems to be acceptable.

6. What is the new methods for the software.

7. I did not get the idea of SAPPHIRE. The authors should try to write the methods and derivation. Without clear methodology, I do not know whether their results are stable.

8. Is it possible to conduct simulation studies, for example, using COSI software?

9. Is the phasing only for SNPs, or it allows genetic variables beyond SNPs, for example multiple alleles, copy numbers, etc. The authors do need to describe their methods in a clear way. Currently, it seems like a black box, which makes evaluation very difficult.

Minor Comments

1. I think the manuscript will improves a lot if the authors can write down the methods used in detail.

Reviewer #3: The review has been uploaded as an attachment.

**Have all data underlying the figures and results presented in the manuscript been provided?**

Reviewer #1: **No: **Lack of supplementary tables for some numerical results; no data availability section

Reviewer #2: Yes

Reviewer #3: None

PLOS authors have the option to publish the peer review history of their article (what does this mean?). If published, this will include your full peer review and any attached files.

Reviewer #1: No

Reviewer #2: No

Reviewer #3: No

---

## [Decision Letter · Decision Letter 1]

11 Jun 2024

Dear Dr Wertenbroek,

We are pleased to inform you that your manuscript entitled "Improving population scale statistical phasing with whole-genome sequencing data" has been editorially accepted for publication in PLOS Genetics. Congratulations!

Yours sincerely,

Yun Li

Academic Editor

PLOS Genetics

Michael Epstein

Section Editor

PLOS Genetics

Comments from the reviewers (if applicable):

Reviewer's Responses to Questions

**Comments to the Authors:**

Reviewer #1: The authors have fairly addressed my previous comments. I have no other concerns.

Reviewer #2: All my comments have been well addressed.

Reviewer #3: I appreciate the authors' work for revising the manuscript. The authors addressed all my comments. I don't have any further comments on the revision. So I think after a final check of format, the manuscript is ready for publication.

**Have all data underlying the figures and results presented in the manuscript been provided?**

Reviewer #1: Yes

Reviewer #2: Yes

Reviewer #3: None

PLOS authors have the option to publish the peer review history of their article (what does this mean?). If published, this will include your full peer review and any attached files.

Reviewer #1: No

Reviewer #2: No

Reviewer #3: No

**Data Deposition**

http://datadryad.org/submit?journalID=pgenetics&manu=PGENETICS-D-23-01356R1

**Press Queries**

---

## [Editor Report · Acceptance letter]

27 Jun 2024

PGENETICS-D-23-01356R1 

Improving population scale statistical phasing with whole-genome sequencing data 

Dear Dr Wertenbroek, 

We are pleased to inform you that your manuscript entitled "Improving population scale statistical phasing with whole-genome sequencing data" has been formally accepted for publication in PLOS Genetics! Your manuscript is now with our production department and you will be notified of the publication date in due course.

With kind regards,

Jazmin Toth

PLOS Genetics

On behalf of:
